# Skeletal Muscle Quality and Quantity Affect Prognosis after Neoadjuvant Chemotherapy with a Triple Regimen of Docetaxel/Cisplatin/5-FU in Patients with Esophageal Cancer

**DOI:** 10.3390/jcm12216738

**Published:** 2023-10-25

**Authors:** Nobuhito Ito, Masahiro Tajika, Tsutomu Tanaka, Keisaku Yamada, Akihiro Takagi, Sachiyo Onishi, Tetsuya Abe, Eiji Higaki, Hironori Fujieda, Yoshitaka Inaba, Kei Muro, Hiroki Kawashima, Yasumasa Niwa

**Affiliations:** 1Department of Endoscopy, Aichi Cancer Center Hospital, 1-1 Kanokoden, Chikusa-ku, Nagoya 464-8681, Japan; no.ito@aichi-cc.jp (N.I.); tstanaka@aichi-cc.jp (T.T.); k.yamada@aichi-cc.jp (K.Y.); ak.takagi@aichi-cc.jp (A.T.); soonishi@aichi-cc.jp (S.O.); yniwa@aichi-cc.jp (Y.N.); 2Department of Gastroenterological Surgery, Aichi Cancer Center Hospital, 1-1 Kanokoden, Chikusa-ku, Nagoya 464-8681, Japan; tabe@aichi-cc.jp (T.A.); ehigaki@aichi-cc.jp (E.H.); h.fujieda@aichi-cc.jp (H.F.); 3Department of Diagnostic and Interventional Radiology, Aichi Cancer Center Hospital, 1-1 Kanokoden, Chikusa-ku, Nagoya 464-8681, Japan; 105824@aichi-cc.jp; 4Department of Clinical Oncology, Aichi Cancer Center Hospital, 1-1 Kanokoden, Chikusa-ku, Nagoya 464-8681, Japan; kmuro@aichi-cc.jp; 5Department of Gastroenterology and Hepatology, Nagoya University Graduate School of Medicine, Tsurumai-cho 65, Showa-ku, Nagoya 466-8550, Japan; h-kawa@med.nagoya-u.ac.jp

**Keywords:** esophageal squamous cell carcinoma, neoadjuvant chemotherapy, sarcopenia

## Abstract

The purpose of this study was to identify factors associated with the prognosis after docetaxel, cisplatin, and 5-fluorouracil (DCF) neoadjuvant chemotherapy (NAC) in patients with advanced esophageal squamous cell carcinoma (ESCC) undergoing surgical resection. We retrospectively examined a total of 100 patients who received neoadjuvant DCF therapy for ESCC at our institution between 2011 and 2020. The psoas muscle index (PMI) was calculated from the psoas muscle area at the L3 vertebral level, and the intramuscular adipose tissue content (IMAC) was calculated from the mean CT value of the multifidus muscle and from four points of subcutaneous fat. The median PMI value was 6.11 cm^2^/m^2^ (range, 3.12–11.07 cm^2^/m^2^) in men and 3.65 cm^2^/m^2^ (range, 2.70–6.82 cm^2^/m^2^) in women. The median IMAC was −0.426 (range, −0.079–−0.968) in men and −0.359 (range, −0.079–−0.671) in women. Based on the PMI, IMAC, and other patient factors, factors associated with NAC-DCF postoperative survival were identified using multivariate Cox regression analysis. A high IMAC was significantly related to overall survival after surgery (*p* = 0.005, hazard ratio 2.699). A comparison of Kaplan–Meier curves showed that the 5-year survival rate was 76.5% in the low IMAC group and 42.7% in the high IMAC group (log-rank test; *p* = 0.001). A low IMAC was associated with good survival outcomes and was an independent prognostic factor in patients with cStage II/III ESCC who were treated with the NAC-DCF regimen and underwent surgical resection.

## 1. Introduction

Esophageal cancer is the tenth most common cause of morbidity and the sixth leading cause of cancer-related death worldwide [1]. According to Japanese guidelines, the treatment of locally advanced esophageal squamous cell carcinoma (ESCC) involves neoadjuvant chemotherapy (NAC) followed by surgery for a radical cure. For many years, NAC with cisplatin and 5-fluorouracil (CF) followed by surgery has been considered the most appropriate treatment for locally advanced ESCC [2]. To further improve the outcomes of patients receiving NAC therapy for ESCC, several clinical trials have been conducted. A phase III trial showing significantly improved survival with the addition of docetaxel to CF in patients with unresectable head and neck cancer [3] led to the development of docetaxel/cisplatin/5-FU (DCF) therapy for esophageal cancer patients as well. A phase II trial by Hara et al. demonstrated the safety of NAC-DCF treatment of esophageal cancer [4]. A multicenter randomized phase II trial conducted by Yamasaki et al. [5] showed that DCF treatment was superior to CF plus adriamycin in terms of recurrence-free survival. Furthermore, the Japanese Clinical Oncology Group (JCOG) conducted the JCOG1109 NExT study, a three-arm phase III trial to evaluate the superiority of DCF versus chemoradiation plus CF versus CF alone as a preoperative therapy [6]. The results demonstrated the superiority of DCF therapy, establishing it as the standard NAC treatment for ESCC in Japan. However, more intensive chemotherapeutic regimens are frequently associated with a variety of adverse effects that may lead to nutritional status deterioration, as typified by sarcopenia.

For decades, the relationship between sarcopenia and various diseases has been reported [7,8], and the psoas muscle index (PMI), calculated from computed tomography (CT) images, has been used as an indicator of sarcopenia [9]. Preoperative sarcopenia has also been identified as a factor that decreases short- and long-term postoperative prognosis after various gastrointestinal cancer surgeries [10,11,12,13]. In patients with stage II or III ESCC, preoperative sarcopenia has been reported to be associated with increased perioperative complications [14,15,16] and poor prognosis [17,18,19]. Recently, not only muscle mass but also muscle quality has been assessed, and the intramuscular adipose tissue content (IMAC) [20] has been used as a measure of muscle quality [21,22]. The association between NAC-DCF and sarcopenia, particularly muscle quality and factors associated with prognosis after NAC-DCF, has not been reported, although NAC-DCF, a more intensive regimen, has become the standard of care. Therefore, this study aimed to identify factors associated with the prognostic value of NAC-DCF in patients with advanced ESCC undergoing surgical resection.

## 2. Materials and Methods

### 2.1. Study Population

A retrospective cohort study comprising 111 consecutive patients diagnosed with cStage II/III ESCC who were treated with the NAC-DCF regimen and underwent surgical resection was conducted at Aichi Cancer Center Hospital, Nagoya, Japan, between January 2011 and December 2020. Eleven of the eligible patients were excluded: eight patients did not have a simple CT scan available before NAC, and analysis was not possible for three patients after lumbar spine surgery or barium examination. The staging was based on the Guidelines for Diagnosis and Treatment of Carcinoma of the Esophagus 2017 by the Japan Esophageal Society [23]. Our study was approved by the Ethics Review Committee of the Aichi Cancer Center (IR031079 2021/09/15) and was conducted per the 1975 Declaration of Helsinki guidelines, as revised in 1983.

### 2.2. Neoadjuvant Chemotherapy

All patients were treated with the DCF regimen (docetaxel, 5-fluorouracil, and cisplatin). The DCF regimen was repeated three times every three weeks as follows: docetaxel at 70 mg/m^2^ given as a 1-h intravenous infusion on day 1 of each cycle, cisplatin at 70 mg/m^2^ as a 2-h intravenous infusion on day 1 of each cycle, and 5-fluorouracil at 750 mg/m^2^ as a continuous infusion on days 1–5. Details of the DCF regimen are presented in Figure 1. The doses used in this study were those used in the JCOG1109 NExT study. Under conditions of tumor progression or chemotherapeutic adverse events, each regimen was performed below the scheduled dose (incomplete cases). All patients received radical subtotal esophagectomy with either two- or three-field lymphadenectomy at least 3–4 weeks after the completion of NAC.

### 2.3. Assessment of Chemotherapy-Related Toxicity and Postoperative Complications

The Common Terminology Criteria for Adverse Events version 5.0 was used to evaluate adverse events. The Clavien–Dindo classification [24] was used to assess postoperative complications, with grade 2 defined as complications from medical therapy, including blood transfusions and central venous nutrition; grade 3 as complications requiring surgical, endoscopic, or interventional radiology (IVR) treatment; and grade 4 as life-threatening complications requiring ICU management.

### 2.4. Measurements of PMI and IMAC

The PMI and IMAC were measured using short-axis slices of pretreatment CT images at the third lumbar vertebral level using SYNAPSE VINCENT software (FUJIFILM Medical Systems, Tokyo, Japan). The value calculated by dividing the psoas muscle area by the square of the value for height was determined as follows: PMI = [(cross-sectional area of bilateral psoas muscle)/(height)^2^(cm^2^/m^2^)] [9]. For the measurement of skeletal muscle quality using CT values, bilateral multifidus muscles were traced at the third lumbar level (the same level at which the psoas muscle cross-sectional area was measured), and the mean CT value was calculated for this site. In addition, subcutaneous fat was traced at four sites at the same level, and the mean CT value was calculated. The mean CT value of the multifidus muscle was divided by the mean CT value of the subcutaneous fat at the four locations, and the value was calculated as follows: IMAC = [mean CT value of bilateral multifidus muscle (HU)/mean CT value of four points of subcutaneous fat (HU)] [9].

### 2.5. Patient Data

Clinical data for all patients were collected from a database prospectively maintained at our institution. Height, weight, and BMI were recorded at first admission. Laboratory data were collected at the first visit to our institution. The general condition at the time of initial admission was assessed using the American Society of Anesthetists (ASA) Physical Status (ASA-PS) score. The prognostic nutrition index (PNI) [25] was calculated using laboratory data as follows: PNI = (10 × serum albumin level [g/dL] + 0.005 × total lymphocyte count [/mm^3^]). Mortality data were collected by a hospital coding system and by contacting the treating practitioner of the patient. Mortality data were determined from the date of the first hospitalization until death or the censor date of the study.

### 2.6. Candidates for Factors Associated with Postoperative Survival of NAC-DCF Patients

Candidate factors associated with NAC-DCF postoperative survival included the patient characteristics of age, sex, BMI, PNI, and ASA-PS score and the sarcopenia-related factors of PMI, IMAC, and degree of tumor progression (cT, cN). The cutoff values were as follows: age of 65 years, ASA-PS score of 2, BMI of 22, PMI of 6.36 (cm^2^/m^2^) in males and 3.92 (cm^2^/m^2^) in females [9], and IMAC of −0.375 in males and −0.216 in females. Patients with an IMAC > each cutoff value and an IMAC < each cutoff value were defined as the high and low IMAC groups, respectively.

### 2.7. Statistical Analysis

SPSS Statistics software version 27.0 (IBM Japan Ltd., Tokyo, Japan) was used for all analyses. Continuous variables are expressed as medians and means, and the differences between medians were analyzed nonparametrically using the Mann–Whitney U test. Categorical variables are indicated by the number of patients, and between-group distribution differences were analyzed using Fisher’s exact test. Cox regression analysis was used to select prognostic factors for NAC-DCF postoperative survival in the patients. Specifically, factors with *p* < 0.1 in the univariate analysis were entered into the multivariate model, and factors with *p* < 0.05 in the multivariate model were selected as prognostic factors. Survival curves were estimated using the Kaplan–Meier method, and comparisons were made using the log-rank test.

## 3. Results

### 3.1. Patient and Treatment Characteristics

Patient and treatment characteristics are shown in Table 1. The median patient age was 65 years (range: 45–79 years), and 77 of 100 patients were male (77%). The median PNI was 45.9 (range: 33.3–57.6). Most cases were cT3 and were accompanied by lymph node metastasis. NAC-DCF in 56 patients was associated with decreased grade 4 neutrophil counts. Clavien–Dindo classification grades 3 and 4 were observed in 37 cases.

### 3.2. Measurement of the PMI and IMAC

The PMI was calculated from the psoas muscle area at the L3 vertebral level, and IMAC was calculated from the mean CT value of the multifidus muscle and of four points of subcutaneous fat (Figure 2). The median PMI was 6.11 cm^2^/m^2^ (range, 3.12–11.07 cm^2^/m^2^) in males and 3.65 cm^2^/m^2^ (range, 2.70–6.82 cm^2^/m^2^) in females. The median IMAC was −0.426 (range, −0.079–−0.968) in males and −0.359 (range, −0.079–−0.671) in females (Table 2).

### 3.3. Factors Associated with Postoperative Survival of NAC-DCF Patients

Univariate Cox regression analysis of nine factors indicated that age over 65 and a high IMAC were candidate risk factors. Multivariate Cox regression analysis revealed that a high IMAC was significantly related to death after surgery (*p* = 0.005, 95% CI 1.343–5.424, hazard ratio 2.699) (Table 3).

### 3.4. Overall Survival of All Patients and Those in the High and Low IMAC Groups

The overall survival rate for all 100 patients was estimated using the Kaplan–Meier method, as shown in Figure 3. The 5-year survival rate was 68.0%.

The survival curves for the high and low IMAC groups are shown in Figure 4. The 5-year survival rate was 76.5% in the low IMAC group and 42.7% in the high IMAC group. The survival rate for the low IMAC group was significantly better than that for the high IMAC group (log-rank test; *p* = 0.001). The patient background for each IMAC group is shown in Table 4. There were no differences in clinical stage, invasion depth, lymph node metastasis, or other tumor diagnoses between the groups. Although there was no significant difference in total number of surgery-related complications (≥grade 3), two or more surgery-related complications (≥grade 3) were frequent in the high IMAC group (*p* = 0.028).

## 4. Discussion

We investigated the association between NAC-DCF and sarcopenia, particularly muscle quality, and factors associated with prognosis after NAC-DCF in patients with ESCC undergoing surgical resection. In the univariate and multivariate analyses, a high IMAC was an independent factor related to survival after surgery. As confirmed using the Kaplan–Meier method, patients in the low IMAC group had a significantly better prognosis than those in the high IMAC group. In other words, the finding of the present study was that loss of skeletal muscle quality (high IMAC) before preoperative chemotherapy impacted overall survival.

In Japan, surgery followed by NAC plus CF has long been considered the most appropriate treatment for locally advanced ESCC. The addition of docetaxel to the CF regimen has been used to treat squamous cell carcinoma of the esophagus, just as it has been administered to patients with unresectable head and neck cancer [3] and in those with metastatic or unresectable locally recurrent anal squamous cell carcinoma [26]. Various clinical phase II trials have been conducted in Japan to further improve the clinical outcome of NAC in ESCC patients [4,5,27]. Similarly, to further improve the clinical outcome of NAC in patients with ESCC, JCOG conducted a 3-arm phase III trial, JCOG1109, to assess the superiority of DCF over CF and the superiority of chemoradiotherapy plus CF over CF as preoperative therapy [6]. In 2022, the results were reported that preoperative DCF showed superior 3-year OS compared with preoperative CF (HR: 0.68, 95% CI: 0.50–0.92; *p* = 0.006). In contrast, preoperative CF plus radiation (CF-RT) did not show superiority over preoperative CF (HR: 0.84: 0.63–1.12; *p* = 0.12). As to adverse events during preoperative treatment, the frequencies of grade 3 or higher neutropenia/febrile neutropenia (FN)/esophagitis/anorexia were 23.4%/1.0%/1.0%/21.4% for CF, 85.2%/16.3%/1.0%/21.4% for DCF, and 44.5%/4.7%/8.9%/14.7% for CF-RT. On the other hand, the frequencies of perioperative complications were as follows: grade 2 or higher pneumonia/anastomotic leakage/recurrent laryngeal paralysis/wound infection were 10.3%/10.3%/15.1%/8.1% for CF, 9.8%/8.7%/10.4%/6.0% for DCF, and 12.9%/12.4%/9.6%/7.3% for CF-RT. Although the frequencies of neutropenia and FN were higher for DCF, the administration of prophylactic antibiotics reduced the severity of FN, resulting in a completion rate of 84.7% for DCF. The results of the JCOG1109 trial promoted preoperative DCF following surgery to the standard treatment in patients with ESCC in Japan. The detailed mechanisms of DCF therapy are not clear. Docetaxel, a microtubule inhibitor, CDDP, a DNA interstrand cross-linking agent, and 5-FU, an antimetabolite, have different modes of action. It has therefore been reported that the combination of these three drugs may lead to a better prognosis due to synergistic effects and loss of cross-resistance [28].

Although such intensive treatment has become the standard treatment and is now widely used in patients with cStage II and III ESCC, chemotherapy causes a variety of adverse events that can lead to nutritional status deterioration such as sarcopenia. Sarcopenia is associated with NAC-related adverse events [29], tumor responsiveness to NAC [21], postoperative complications [19], and prognosis [19,21,22]. Specifically, previous studies have reported that in patients with ESCC treated with preoperative chemotherapy, a decrease in skeletal muscle mass (especially PMI) before NAC worsens the prognosis after surgery [22] and that a decrease in muscle quality (especially a high IMAC) worsens the prognosis [21,22]. Similarly in this study, two or more surgery-related complications (≥grade 3) were also significantly more common in the high IMAC group. These results may have influenced the worse prognosis in the high IMAC group.

The assessment of skeletal muscle “quality” using the IMAC has recently received considerable attention. The IMAC is calculated as the CT value of the multifidus muscle/CT value of the back subcutaneous fat and is a negative value. Fat degeneration, as measured by the IMAC, has been reported to correlate with aging, muscle weakness, and functional decline [30] and has been considered a factor in sarcopenia in recent years. The utility of the IMAC to predict clinical outcomes has been previously reported in a variety of cancers [20,21,22,31]. Previous studies in esophageal cancer have included patients with esophageal cancer who received CF therapy (fluorouracil + cisplatin) as well as those who received DCF therapy as NAC. In previous studies, the 5-year survival rates for the high and low IMAC groups were 29.9% and 61.9%, indicating a poor prognosis in the high IMAC group. In this study, the 5-year survival rate was 76.5% in the low IMAC group and 42.7% in the high IMAC group. As in previous studies involving types of chemotherapy other than DCF, a high IMAC was identified as an independent factor related to poor prognosis after surgery. This study is the first study to reveal the prognosis of patients treated with NAC-DCF only and the quality of the muscles, especially in patients with sarcopenia. Unexpectedly, the PMI was not identified as an independent factor in the current study. However, this suggests that the IMAC may be a better index for the evaluation of sarcopenia than the PMI. Because the PMI is expressed by measuring the area of the psoas muscle at the L3 level, including the fat, the skeletal muscle area may not accurately reflect the real skeletal muscle condition. Therefore, we believe that the IMAC better reflects a decrease in muscle mass in patients. NAC-DCF therapy is difficult to administer in elderly patients because of the report that this treatment approach does not improve prognosis in patients older than 75 years [32], and a decreased grade 4 neutrophil count occurred in approximately half of the patients in the current study cohort. This may be due in part to patient selection for NAC-DCF, as indicated by the fact that the median age of the cohort in this study was younger than that in other studies. However, in this study, there were no significant differences in patient characteristics between the high and low IMAC groups; in other words, preoperative evaluation of the IMAC is very important because it is difficult to predict whether the IMAC will be high or low before treatment, especially in patients treated with NAC-DCF. Furthermore, preoperative evaluation of the IMAC may provide an opportunity to choose NAC-CF rather than NAC-DCF in high IMAC patients.

The relationship between a high IMAC and poor prognosis suggests the need for intervention to improve skeletal muscle quality. Previous studies suggested a potential benefit of a rehabilitation program during NAC or early exercise during NAC to reduce the risk of skeletal muscle loss [33,34]. Furthermore, an omega-3 fatty acid dietary supplement or an amino acid supplement has been reported to improve sarcopenic conditions [35,36]. However, the effects of these interventions during NAC on the long-term outcomes of patients remain unclear. Further trials are required to clarify whether nutritional intervention and physical exercise improve muscle quality and thereby contribute to prolonging survival.

This study had several limitations. First, this was a single-center, retrospective cohort study of a small number of patients who were cardiopulmonary preserved and able to undergo surgery and thus may be subject to selection bias. Second, this study only included patients with ESCC. Thus, patients with esophageal adenocarcinoma were not included. Third, the cutoff values for the PMI and IMAC were based on -2 SD values of healthy volunteers [9]. Fourth, we could not evaluate physical performance, such as grip strength or gait speed. The Asian Working Group for Sarcopenia [37] has suggested an algorithm for sarcopenia based on measurements of both physical performance and muscle mass. Fifth, the effect of the reduction in intake of food due to esophageal cancer on the IMAC could not be evaluated. Finally, as with previous reports on the IMAC, the mechanism by which it affects prognosis remains unclear. Future prospective studies are needed to determine whether interventions including exercise and nutritional therapy from the start of NAC can prevent loss of muscle quality and improve overall survival.

## 5. Conclusions

A low IMAC was associated with survival outcomes and was an independent prognostic factor in patients with cStage II/III ESCC who were treated with the NAC-DCF regimen and underwent surgical resection.

## Figures and Tables

**Figure 1 jcm-12-06738-f001:**
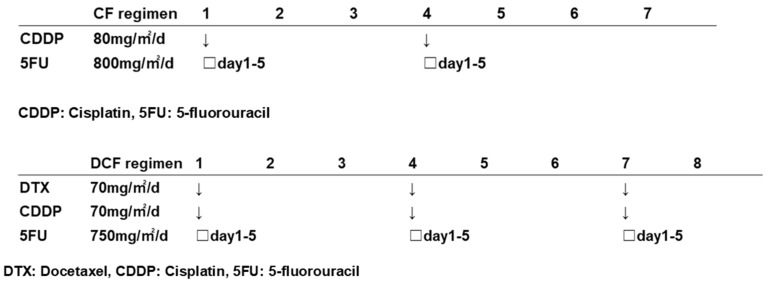
Details of the CF regimen and DCF regimen.

**Figure 2 jcm-12-06738-f002:**
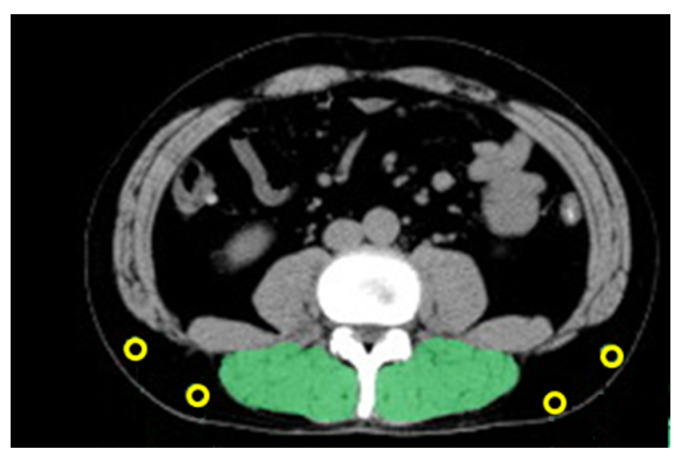
Methods of IMAC measurement. The mean CT value of the multifidus muscle was calculated by green area. The mean CT value of subcutaneous fat was calculated by four small points.

**Figure 3 jcm-12-06738-f003:**
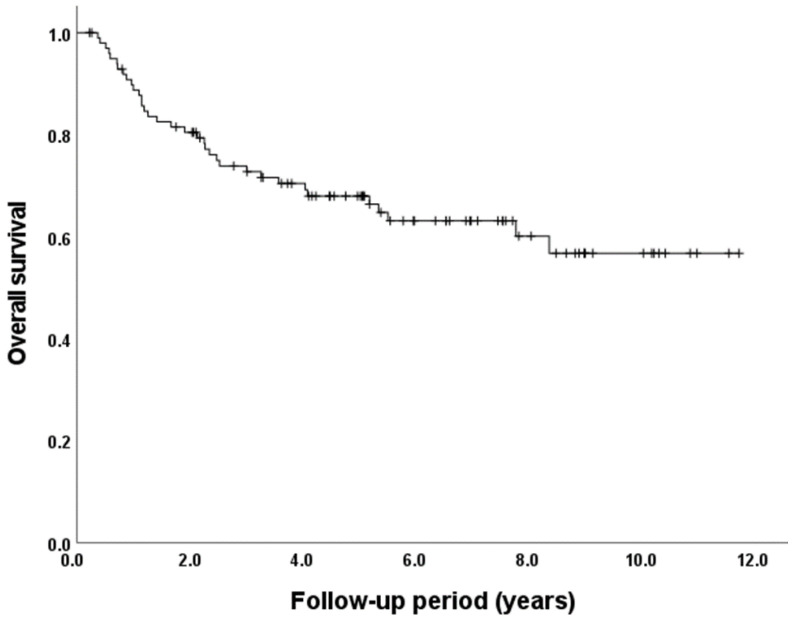
Overall survival of all patients.

**Figure 4 jcm-12-06738-f004:**
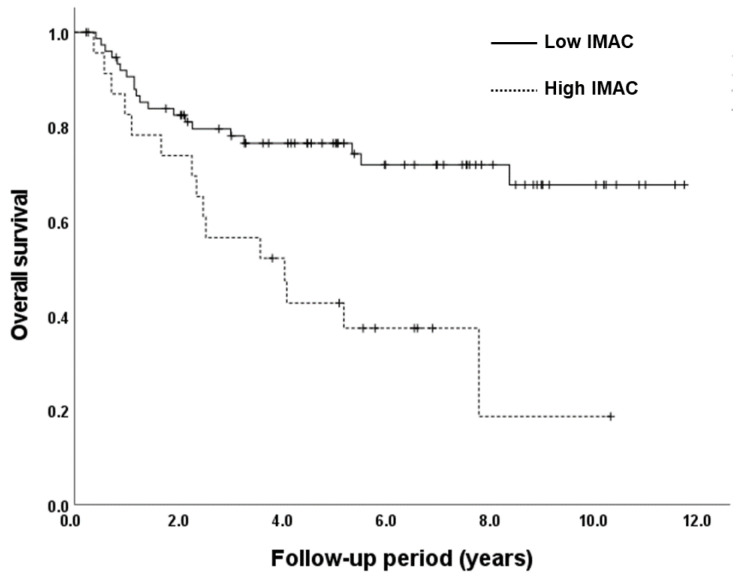
Overall survival of each IMAC group.

**Table 1 jcm-12-06738-t001:** Patient characteristics and treatment outcomes.

Variables	
Age, median (range)	65 (45–79)
Sex	
Male, n (%)	84 (84%)
Female, n (%)	16 (16%)
Body mass index (kg/m^2^), median (range)	21.5 (13.8–27.6)
ASA-PS score (1/2-)	26/74
PNI, median (range)	45.9 (33.3–57.6)
Hb (g/dL), median (range)	11.4 (7.9–14.3)
Location (Ut/Mt/Lt), n	17/46/37
Clinical T (1, 2/3), n	13/87
Clinical N (1, 2/3), n	47/53
Clinical stage (II/III), n	17/83
Neutrophil count decreased grade 4, n (%)	56 (56%)
Operative time, (min), median (range)	472 (321–848)
Intraoperative bleeding (mL), median (range)	230 (20–3190)
Surgery-related complications (≧grade 3), n (%)	37 (37%)

ASA-PS: American Society of Anesthesiologists Physical Status, PNI: prognostic nutrition index.

**Table 2 jcm-12-06738-t002:** Measurement of the PMI and IMAC.

Variables	Male	Female
PMI (cm^2^/m^2^), median (range)	6.11 (3.12–11.07)	3.65 (2.70–6.82)
IMAC, median (range)	−0.426 (−0.968–−0.079)	−0.359 (−0.671–−0.079)

PMI: psoas muscle index, IMAC: intramuscular adipose tissue content.

**Table 3 jcm-12-06738-t003:** Univariate and multivariate COX regression analysis.

Variables	Categorization	Univariate Analysis			Multivariate Analysis		
		HR	95% CI	*p* Value	HR	95% CI	*p* Value
Age	≧65	1.74	0.896–3.381	0.097	1.343	0.670–2.691	0.406
Sex	male	1.125	0.436–2.901	0.807			
BMI	<22	1.294	0.643–2.603	0.47			
PNI	<45	1.17	0.595–2.302	0.649			
PMI	high	0.727	0.366–1.444	0.362			
IMAC	high	2.912	1.483–5.721	0.002	2.699	1.343–5.424	0.005
ASA-PS	≧2	1.97	0.817–4.747	0.131			
cT	III	2.18	0.666–7.139	0.198			
cN	≧2	1.211	0.622–2.357	0.574			

BMI: body mass index, PNI: prognostic nutrition index, PMI: psoas muscle index, IMAC: intramuscular adipose tissue content, ASA-PS: American Society of Anesthesiologists Physical Status.

**Table 4 jcm-12-06738-t004:** Patient characteristics and treatment outcomes stratified by IMAC.

Variables	High IMAC (n = 23)	Low IMAC (n = 77)	*p* Value
Age, median (range)	67 (51–79)	65 (45–79)	0.065
Sex			
Male, n (%)	21 (91%)	63 (82%)	0.349
Female, n (%)	2 (9%)	14 (18%)	
Body mass index (kg/m^2^), median (range)	21.8 (17.3–27.6)	21.3 (13.8–27.1)	0.024
ASA-PS score (1/2-)	19/4	55/22	0.417
PNI, median (range)	47.0 (33.3–51.6)	45.8 (35.0–57.6)	0.67
Hb (g/dL), median (range)	11.5 (8.9–13.7)	11.4 (7.9–14.3)	0.993
Location (Ut/Mt/Lt), n	4/10/9	13/36/28	0.96
Clinical T (1, 2/3), n	3/20	10/67	1
Clinical N (1, 2/3), n	14/9	33/44	0.157
Clinical stage (II/III), n	4/19	13/64	1
Neutrophil count decreased grade 4, n (%)	12 (52%)	44 (57%)	0.811
Operative time, (min), median (range)	452 (353–848)	477 (321–682)	0.534
Intraoperative bleeding, (mL), median (range)	250 (80–3190)	220 (20–1370)	0.555
Surgery-related complications (≧grade3), n (%)	10 (43%)	27 (35%)	0.472
Two or more surgery-related complications (≧grade3), n (%)	5 (22%)	4 (5%)	0.028

ASA-PS: American Society of Anesthesiologists Physical Status, PNI: prognostic nutrition index.

## Data Availability

Not applicable.

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
