# Peer review of "Skeletal Muscle Quality and Quantity Affect Prognosis after Neoadjuvant Chemotherapy with a Triple Regimen of Docetaxel/Cisplatin/5-FU in Patients with Esophageal Cancer"

_jcm, 2023, doi:10.3390/jcm12216738_

Round 1
Reviewer 1 Report
Reviewer Report:
The article titled “Skeletal muscle quality and quantity affect prognosis after neoadjuvant chemotherapy with triple regimens of docetaxel/cis-platin/5-FU in patients with esophageal cancer” by Tajika et al is an interesting article for prognosis of esophageal squamous cell carcinoma (ESCC).
The article is written well and could be published after satisfactory explanations of the following questions.
1. What is the basis for the combination of drugs Docetaxel/cis-Platin/5-Fluorouracil to treat ESCC? What is the added benefit of adding docetaxel for the treatment? The drugs acts by different mechanism and how do you differentiate the efficacy of the drugs? Which drug of the regimen is more efficient? These details need to be included in the manuscript for better understanding of the selection of drug combination.
2. The authors claim by using the described regimen to ESCC patients, the prognosis and 5-year survival rate has improved to 76% in low IMAC patients. But how does the addition of docetaxel influence the survival outcome of patients when combination of cis-Platin/5-Fluorouracil doesn’t improve it considerably? What is new by additional effect of docetaxel drug in the treatment? There is no rational and explanation for the effect of docetaxel in the regimen.
3. A comparison table of CF and DCF in the manuscript will help the readers to understand the effect of different regimens.
4. It is well known that intramuscular adipose tissue content (IMAC) has been used as a measure of muscle quality during the NAC-CF treatment for prognosis of ESCC. How does addition of docetaxel improve the IMAC (low) in the current investigation? What is the mechanism? What is the feedback loop of IMAC content and ESCC treatment?
5. The study directly say’s that the survival rate increased because IMAC is low during DCF regimen but fail to provide any rational explanation for the better prognosis factor. Inclusion of an explanation in the manuscript would strengthen the claim of authors.
6. What is the basis for the described dosage given to the patients (docetaxel at 70 mg/m2 given as a 1-h intravenous infusion on day 1 of each cycle, cisplatin at 70 mg/m2 as a 2-h intravenous infusion on day 1 of each cycle, and 5-fluorouracil at 700 mg) ? 5-F-Uracil is provided in disproportionately large dose? Reasons?
Though adenocarcinoma is closely associated with ESSC but it in not included in the study. Probable reasons to be included because docetaxel is another good drug to treat adenocarcinoma.
Author Response
"Please see the attachment."

Reviewer 2 Report
To be considered in the discussion: Patients were all affected by esophageal cancer and this should be considered as possible bias due to the (poor /low protein?) food intake in these patients and related impact on skeletal muscle “quality” using IMAC.
Dietary and exercise patterns possibly contributing to IMAC should be discussed as well as the importance of an early surgery in esophageal cancer to minimize sarcopenia
Author Response
"Please see the attachment."

Round 2
Reviewer 1 Report
Revision-1 Reviewer Report:
The authors have not answered most of the comments raised during first revisions.
1. The comments such as "What is the added benefit of adding docetaxel for the treatment? The drugs acts by different mechanism and how do you differentiate the efficacy of the drugs? Which drug of the regimen is more efficient? These details need to be included in the manuscript for better understanding of the selection of drug combination" were not answered in the revised manuscript, instead citing that the doses were demonstrated by an another study JCOG1109 NExT. I think the authors don't follow the comments.
2. Similarly, my second comments "
The authors claim by using the described regimen to ESCC patients, the prognosis and 5-year survival rate has improved to 76% in low IMAC patients. But how does the addition of docetaxel influence the survival outcome of patients when combination of cis-Platin/5-Fluorouracil doesn’t improve it considerably? What is new by additional effect of docetaxel drug in the treatment? There is no rational and explanation for the effect of docetaxel in the regimen" were not answered by the authors. There is no explanation for the effect of docetaxel in the regimen.
3. A comparison table of CF and DCF in the manuscript is important to understand the effect of different regimens. This will reveal added benefits of the DCF regimen and I don't agree that this will confuse the scientific readers.
4. The authors are not clear about the mechanism of action of docetaxel addition to the regimen.
I regret to inform that all the comments are not addressed by the authors and the study is incomplete, lacks scientific clarity in explanation and rationalization of the outcomes.
Author Response
Thank you very much for taking the time to review this manuscript, again.
We have corrected the wording of the part you pointed out.
Comments1
The comments such as "What is the added benefit of adding docetaxel for the treatment? The drugs acts by different mechanism and how do you differentiate the efficacy of the drugs? Which drug of the regimen is more efficient? These details need to be included in the manuscript for better understanding of the selection of drug combination" were not answered in the revised manuscript, instead citing that the doses were demonstrated by an another study JCOG1109 NExT. I think the authors don't follow the comments.
Response1.
→Thank you for informative comments. The efficacy of NAC-DCF with docetaxel was reported in 2007 in SCC of the head and neck, and has since been reported in various clinical trials in esophageal cancer as well. Therefore, we have added the history of the development of the regimen to the INTRODUCTION. We have also added a previously reported comparison of survival rates with NAC-CF in the Discussion section.
To further improve the outcomes of patients receiving NAC therapy for ESCC, several clinical trials have been conducted. The Phase III trial of significantly improved survival with the addition of docetaxel to CF in unresectable head and neck cancer[3] have led to the development of docetaxel/cisplatin/5-FU (DCF) therapy in esophageal cancer as well. A Phase II trial by Hara et al. demonstrated the safety of DCF in the NAC treatment of esophageal cancer[4]. A multicenter randomized phase II conducted by yamasaki et al. [5]showed that DCF treatment was superior to CF plus Adriamycin in terms of recurrence-free survival.
In previous studies, the 5-year survival rate for the high and low IMAC groups were 29.9% and 61.9%, resulting in a poor prognosis in the high IMAC group. In this study, the 5-year survival rate was 76.5% in the low IMAC group and 42.7% in the high IMAC group. As in previous studies involving types of chemotherapy other than DCF, high IMAC was identified as an independent factor related to poor prognosis after surgery.
Comments2.
Similarly, my second comments "
The authors claim by using the described regimen to ESCC patients, the prognosis and 5-year survival rate has improved to 76% in low IMAC patients. But how does the addition of docetaxel influence the survival outcome of patients when combination of cis-Platin/5-Fluorouracil doesn’t improve it considerably? What is new by additional effect of docetaxel drug in the treatment? There is no rational and explanation for the effect of docetaxel in the regimen" were not answered by the authors. There is no explanation for the effect of docetaxel in the regimen.
Response2.
We thank you for your helpful comments. We are sorry for our inadequate and confusing description, but it has been reported that high-IMAC is associated with poor prognosis in NAC-CF as well. As mentioned in Response 1, 5-year survival rates in high-IMAC and low-IMAC in previous studies have been added to Discussion.
In previous studies, the 5-year survival rate for the high and low IMAC groups were 29.9% and 61.9%, resulting in a poor prognosis in the high IMAC group. In this study, the 5-year survival rate was 76.5% in the low IMAC group and 42.7% in the high IMAC group. As in previous studies involving types of chemotherapy other than DCF, high IMAC was identified as an independent factor related to poor prognosis after surgery.
Comments3.
A comparison table of CF and DCF in the manuscript is important to understand the effect of different regimens. This will reveal added benefits of the DCF regimen and I don't agree that this will confuse the scientific readers.
Response3.
As you pointed out, we have added a figure comparing the NAC-CF and NAC-DCF regimens to Figure 1 for readers’ deeper understanding of the regimens.
Figure1. Details of the CF regimen and DCF regimen
Comments4. The authors are not clear about the mechanism of action of docetaxel addition to the regimen.
Response4.
Thank you for your helpful comments, although the detailed mechanisms of DCF are not clearly known, we have added the following literature discussion to the Discussion.
The detailed mechanisms of DCF therapy are not clear. Docetaxel, a microtubule inhibitor, CDDP, a DNA interstrand cross‐linking agent, and 5‐FU, an antimetabolite, have different modes of action. It has therefore been reported that the combination of these three drugs may lead to a better prognosis due to synergistic effects and loss of cross-resistance[27].

Round 3
Reviewer 1 Report
The manuscript can be accepted